## RESEARCH ARTICLE

# Identifying determinants of readmission and death post-stroke using explainable machine learning

Emir Veledar[1]*, Lili Zhou[1], Omar Veledar[2], Hannah Gardener[1], Carolina M. Gutierrez[1], Scott C. Brown[1], Farya Fakoori[1], Karlon H. Johnson[1], Victor J. Del Brutto[1], Ayham Alkhachroum[1], David Z. Rose[3], Gillian Gordon Perue[1], Negar Asdaghi[1], Jose G. Romano[1], Tatjana Rundek[1]

1 University of Miami Miller School of Medicine, Miami, Florida, United States of America, 2 Beevadoo e.U, Graz, Austria, 3 Department of Neurology, University of South Florida College of Medicine, Tampa, Florida, United States of America

* exv439@miami.edu

## Abstract

### Background

Stroke remains a global health challenge with high rates of mortality and rehospitalization placing significant demands on healthcare systems. Identifying factors that determine outcomes of post-hospitalization improves resource allocation. Traditional statistical prediction models are suboptimal for the analysis of complex, multi-dimensional datasets. The objective of our study is to define the extended list of clinical and non-clinical predictors, which we believe can be achieved using Explainable Machine Learning (XML) models as an expansion of conventional methods.

### Methods

We evaluated 11 established XML models that represent key ML methodologies to predict 90-day outcomes, namely mortality and rehospitalization among stroke survivors. The study population are 1,300 post-stroke individuals enrolled in the Transitions of Care Stroke Disparities Study (TCSD-S) (NIH/NIMH, NCT03452813) between June 2018 – October 2022. The care after transition data is sourced from participating comprehensive stroke centers and from the Florida Stroke Registry. The analysis incorporated clinical (e.g., age, stroke severity, comorbidities) and non-clinical factors including Social Drivers of Health (SDOH). A combined ranking approach, using Weighted Importance Scores and Frequency Counts, identified significant predictors across models.

**Data availability statement:** The FSR uses data from Get with The Guidelines-Stroke® (GWTG-S). Due to data-sharing agreements, researchers must apply for access at http://www.heart.org/qualityresearch, with proposals reviewed by GWTG-S and FSR committees upon reasonable request.

**Funding:** The author(s) received no specific funding for this work.

**Competing interests:** The authors have declared that no competing interests exist.

## Results

The resulting list of selected predictors included both established clinical factors and non-clinical factors, which enhanced prediction accuracy. Out of 38 identified predictors, 20 are non-clinical variables reflecting the importance of SDOH, environmental factors, and behavioral modifications beyond traditional clinical predictors of death/readmission. A secondary analysis restricted to ischemic stroke patients (n = 1,038) yielded virtually identical predictive performance, indicating robustness of the model within this subgroup.

## Conclusions

Integrating SDOH, environmental factors, and behavioral modifications alongside traditional clinical predictors enhances the predictive accuracy of post-stroke outcome models. This underscores the critical role of addressing socioeconomic disparities during post-stroke transitions of care. Moreover, XML models' ability to identify predictors spanning clinical and non-clinical domains suggests their potential to guide recovery. The resulting predictors are crucial for post-hospital care and hold strong potential for identifying individuals at risk of stroke, making them potentially significant across pre-stroke and hospitalization stages.

## 1. Introduction

Stroke challenges healthcare systems across the entire continuum of care. As the third leading cause of mortality and a major source of disability worldwide [1], stroke is a major chronic non-communicable condition associated with reduced population well-being [2]. Assessing stroke risk factors is a crucial initial step in decreasing its burden. Predictive models can be applied at three stages: primary prevention, response to acute treatment, and post-stroke outcomes including post-hospitalization recurrence and readmissions. The complexity of stroke adjudication, diagnosis, and treatment algorithms often restricts model effectiveness, demanding phase-specific approaches. Current meta-analyses reveal that no high-quality predictive or explanatory models exclusively addressing stroke risk currently exist [3,4]. Existing models often rely on non-modifiable risk factors and borrow heavily from frameworks developed for composite cardiovascular diseases. For the general population, all measures of model quality remain low, highlighting significant gaps in predictive accuracy and reliability. While recent studies have not substantially improved prediction intervals, they have introduced greater inclusivity by incorporating non-traditional biomarkers, such as genetic and polygenic scores, and leveraging novel ML methods, marking incremental progress in methodology and scope. Our focus is on developing and improving models specifically for the rehospitalization and death.

In the first (primary prevention) phase, predictive models aim to assess stroke risk within the general population but face substantial limitations, with most models achieving ROC scores below 0.7. These models rely on cardiovascular markers such

as the AHA Life's Essential 8 [5] which lack stroke specificity. Adding stroke-specific indicators is typically used for high-risk individuals to enhance prediction accuracy but is impractical for general screening [6].

The second phase focuses on patients receiving hospital care for acute stroke. Here, predictive power increases with detailed clinical data from advanced diagnostics like, brain and head/neck vessel imaging enabling the use of ML models to refine treatment. However, the wide variation in stroke subtypes and specific predictors for each subtype limits the development of a universal model, necessitating predictor-specific models for different stroke etiologies.

In the post-hospitalization stroke period, stroke survivors face heightened risks of complications, recurrence, and rehospitalization. Previous studies have the prevalence of stroke survivors discharged home ranging between 43–92% [7–12], often requiring ongoing care, rehabilitation, and adherence to treatment plans. A vast number of studies have focused on analyzing individual medical characteristics of stroke patients. However, our research takes a different approach, concentrating on additional variables that capture broader, non-individual characteristics. This study aims to predict 90-day post-discharge outcomes from acute stroke hospitalization by identifying key predictors of mortality and readmission. By leveraging data from multiple sources, the study employs XML models, which integrate clinical factors, SDOH, and health behaviors. Using multiple XML models, we aim to isolate robust, modifiable predictors of 90-day outcomes of death or readmission post-stroke. Our approach underscores XML's ability to capture complex, non-linear relationships, enhancing interpretability and trust in predictive insights to inform post-stroke care. Therefore, our main research question targets to determine how XML models improve the prediction of 90-day mortality and readmission outcomes following acute stroke hospitalization even when the sample size is small and the design is unbalanced.

## 2. Materials and methods

We adopt the Weighted Importance Score and Frequency Count (WISFC) framework for aggregating feature-importance across multiple models, as recently detailed [13]. This approach systematically combines magnitude and consistency information from diverse explainers to produce a robust, consensus-based ranking of predictors.

### 2.1. Study population

Our study population is comprised of 1,300 post-stroke individuals enrolled in the Transitions of Care Stroke Disparities Study (TCSD-S) (NIH/NIMH, NCT03452813) between June 2018-October 2022 combined by information from participating comprehensive stroke centers from the Florida Stroke Registry [14]. The TCSD-S is an observational study of stroke survivors investigating factors that influence successful transitions post-stroke. Eligible participants were adults aged 18 years and older, diagnosed with either acute ischemic stroke or intracerebral hemorrhage, and discharged to either a rehabilitation facility or directly home. Although ischemic and intracerebral hemorrhage (ICH) strokes differ pathophysiologically, in our cohort, both groups had relatively low Modified Rankin Scale scores at discharge and showed no statistically significant difference in 90-day outcome rates. Given this similarity, and to preserve sample size and generalisability, both stroke types were retained in the analysis. Hospital care coordinators conducted interviews at discharge to assess SDOH. Follow-up structured interviews at 30 and 90 days post-discharge tracked readmissions, emergency room visits, discharge education, and behavioral modifications. The TCSD-S enrollee data from 10 collaborating comprehensive stroke centers were linked to the American Heart Association's Get With The Guidelines–Stroke (GWTG-S) Database, providing additional clinical and demographic information such as race/ethnicity, sex, age, insurance status, stroke severity, and pre-stroke health conditions. Patients who died within 30 days or had specific post-discharge dispositions (e.g., transferred to hospice care) were excluded from the study.

Only those patients who provided written informed consent participated in the TCSD-S. The study protocol was approved by the Institutional Review Board of the University of Miami Protocol ID20170892.

## 2.2. Study variables

The primary outcome variable was a composite of hospital readmission or mortality within 90 days after discharge from the index hospitalization. Since approximately half of all readmissions within one year after a stroke occur within the first 90 days [15–17], predicting 90-day outcomes is useful.

Independent variables were extracted from three sources: the TCSD-S which provides insight into individual SDOH, the GWTG-Stroke database provides clinical and index stroke characteristics, and the Social Contextual Indicators for Research and Analysis (SCIERA) database [18]. The database provides neighborhood-level SDOH factors (including detailed zip code characteristics), offering a broader context for individual patient data.

We included nine comprehensive arrays of potential predictors encompassing neighborhood socio-economic, and demographic characteristics, individual social determinants of health, health characteristics before the stroke, index stroke characteristics, acute care variables, hospital characteristics, discharge status, and measures of Adequate Transition of Care [19]:

1. Patients' sociodemographic Characteristics: Age, sex, race/ethnicity, and type of insurance.

2. Individual SDOH: Language spoken at home, education status, prior work status, difficulty paying for medical care, difficulty paying for necessities (e.g., food, electricity), living arrangements, and social support [20].

3. Health Characteristics Before the Stroke: Prior ambulation status, diabetes, hypertension, dyslipidemia, atrial fibrillation, peripheral vascular disease, coronary artery disease, previous stroke/Transient Ischemic Attack (TIA), carotid stenosis, chronic renal insufficiency, sleep apnea, depression, smoking status, drug/alcohol use, overweight/obesity, and history of deep vein thrombosis/pulmonary embolism (DVT/PE).

4. Index Stroke Characteristics: Stroke etiology, National Institutes of Health Stroke Scale (NIHSS) score, final clinical diagnosis related to stroke, presence of weakness/paresis, altered level of consciousness, aphasia/language disturbance, other neurological signs/symptoms, mode of arrival (e.g., ambulance, self-transport), and length of hospital stay.

5. Acute Care Variables: intravenous thrombolysis, endovascular therapy, discharge medications including antiplatelet agents, anticoagulants, and statins, provision of defect-free care, and presence of active bacterial or viral infection at admission or during hospitalization.

6. Hospital Characteristics: Stroke center type (e.g., primary stroke center, comprehensive stroke center), teaching hospital status, and the number of beds.

7. Discharge Status: Modified Rankin Scale (mRS) score at discharge, discharge disposition (e.g., home, rehabilitation facility), and ambulation status at discharge.

8. Neighborhood (Zip-Code) Characteristics: Percentages of Hispanic, Non- Hispanic Black, and Non-Hispanic White residents; percentage of residents with a bachelor's degree; median household income; percentage of high school completion; rural-urban commuting area codes; total housing population; the ratio of owner-occupied housing; housing density [21]; percentage below the poverty line; unemployment rate; densities of tobacco, alcohol, restaurant, fast food, grocery, pharmacy, and gym businesses; and counts of hospitals, clinics, and rehabilitation centers in the area. Crowding, in our context, is a measure of population density within a zip code. It is calculated by dividing the total housing population by the adjusted number of housing units, which is determined by dividing the count of housing units by the median number of rooms per unit. This metric provides insight into how densely populated a given area is concerning its housing capacity. Total Housing Population is defined as the total number of people living in owner or renter-occupied housing in a zip code. Social Support Size is the measure of how many persons a patient knows that they feel close to (i.e., persons they can talk to or reach out to if needed with 3 categories: "None", "1-2", "3 or more").

9. Adequate Transition of Care (ATOC): Adherence to at least 75% of applicable transition of care behavior modifications, including filled medications and taken as prescribed 90–100% of the time; attending outpatient therapy or attended and completed therapy if prescribed; has seen a medical provider after discharge; stopped using tobacco, alcohol, marihuana, and other drugs; exercising by regular walking on a treadmill or outside, or regular exercise other than walking; modified diet per recommendation after stroke.

The methods for obtaining outcome data and adjudicating endpoints have been detailed in the prior publication [19].

## 2.3. XML techniques and implementation

To predict post-discharge stroke outcomes, we applied a comprehensive array of XML techniques aimed at identifying the most important predictive variables from a pool of diverse clinical, socioeconomic, and behavioral data detailed above. This section outlines the methodology used to handle the 73 potential variables, the application of 11 different XML models, and the process of ranking variable importance.

**Step 1: Identifying and Selecting Variables**: The 73 study variables were selected as they represent strong candidates based on prior evidence [14,19,22] and their availability in the dataset. As detailed in the previous section, they were drawn directly from multiple data sources and span key domains such as sociodemographic characteristics, social determinants of health, pre-stroke health status, stroke-specific and acute care data, hospital and discharge characteristics, neighborhood factors, and transition of care adequacy. By incorporating these diverse variables, we ensured a comprehensive basis for modeling post-stroke recovery and readmission risk.

**Step 2: Applying XML Methods**: To process these 73 variables, we employed 11 different XML models. We use logistic regression as a universal benchmark, alongside 10 additional models encompassing regression-based, tree-based, and distance-based algorithms. These 10 models represent state-of-the-art approaches widely accepted in ML, ensuring comprehensive coverage of predictive methodologies. This method uses one data set randomly divided into 10 parts. Nine of those parts are used for training and a tenth for testing. This procedure is repeated 10 times reserving a different tenth for testing [23]. Each XML model training was carried out under 10-fold cross-validation to optimize hyperparameters and estimate out-of-sample performance. All continuous predictors in the training set were then mean-centered and scaled to unit variance; the same centering and scaling parameters were subsequently applied to the test set to avoid data leakage. No additional feature-engineering transformations were performed beyond this normalization step.

All candidate predictors were assessed for completeness prior to model development. Patients with missing values in any of the predictors listed in Table 1 were excluded from the analysis; no imputation methods were applied. Of the original cohort of 1,200 stroke patients, 73 (6.1%) were removed due to incomplete records, yielding a final sample of 1,127 patients with fully observed data.

To assess whether model performance differed when restricted to ischemic stroke, we performed the secondary analysis in ischemic stroke subset. We excluded the 89 hemorrhagic cases from our final cohort of 1,127 patients, yielding 1,038 ischemic strokes (92.1%). Predictors specific to stroke subtype (variables 16 and 38) were omitted. Discrimination (AUC) and calibration metrics remained virtually unchanged compared to the primary analysis.

All models were tuned or set under a uniform 10-fold cross-validation framework. For the penalized regressions (LASSO, Ridge, and Elastic Net with $\alpha = 0.5$), we selected the smallest $\lambda$ that minimized cross-validated error in each fold. Principal Component Regression was run with a fixed 10 components ($k = 10$). The k-Nearest Neighbors model evaluated 20 candidate k values via cross-validation. Support Vector Machines employed the radial basis kernel with default cost and $\gamma$ settings. Random Forests were grown with 500 trees and the default mtry. Gradient Boosting was configured with 500 trees, interaction depth = 3, and shrinkage = 0.1. Finally, XGBoost models were trained for up to 100 rounds (with early stopping after 10 rounds), max_depth = 3, $\eta = 0.3$, subsample = 0.8, and colsample_bytree = 0.7.

**Table 1. Participants' demographic and stroke characteristics – selected noteworthy variables.**

| Selected Variables | Total (N = 1300) | Death or readmission in 90 days | | P Value |
| --- | --- | --- | --- | --- |
| | | NO (N = 1094) | YES (N = 206) | |
| Age, mean (SD) | 63.8 (13.9) | 64.0 (13.9) | 62.7 (14.3) | 0.23 |
| Male sex | 732 (56.3%) | 621 (56.8%) | 111 (53.9%) | 0.44 |
| Ischemic stroke | 1196 (92.0%) | 1011 (92.4%) | 185 (89.8%) | 0.21 |
| ICH | 104 (8.0%) | 83 (7.6%) | 21 (10.2%) | |
| NIH Stroke Scale, median (range) | 3 (0–33) | 2 (0–33) | 4 (0–28) | <0.001 |
| mRS baseline, median (range) | 1 (0–5) | 1 (0–5) | 2 (0–5) | <0.001 |
| Length of Stay, median (range) | 4.1 (0.7–66.3) | 4.0 (0.7–41.8) | 4.5 (1.1–66.3) | <0.001 |
| Diabetes Mellitus | 411 (31.6%) | 332 (30.4%) | 79 (38.4%) | 0.023 |
| CAD or prior MI | 242 (18.6%) | 192 (17.6%) | 50 (24.3%) | 0.023 |
| Previous Stroke or TIA | 270 (20.8%) | 207 (18.9%) | 63 (30.6%) | <0.001 |
| Chronic Renal Insufficiency | 98 (7.5%) | 65 (5.9%) | 33 (16.0%) | <0.001 |
| ATOC | 832 (64.0%) | 728 (66.5%) | 104 (50.5%) | <0.001 |
| Carotid Stenosis | 56 (4.3%) | 44 (4.0%) | 12 (5.8%) | 0.24 |
| Non-Hispanic White race | 665 (51.2%) | 564 (51.6%) | 101 (49.0%) | 0.2 |
| Non-Hispanic Black race | 296 (22.8%) | 244 (22.3%) | 52 (25.2%) | |
| Hispanic | 286 (22.0%) | 246 (22.5%) | 40 (19.4%) | |
| Other | 53 (4.1%) | 40 (3.7%) | 13 (6.3%) | |
| Private Insurance | 291 (22.4%) | 256 (23.4%) | 35 (17.0%) | 0.046 |
| Medicare | 580 (44.6%) | 493 (45.1%) | 87 (42.2%) | |
| Medicaid | 63 (4.9%) | 51 (4.7%) | 12 (5.8%) | |
| Self/No Insurance | 366 (28.2%) | 294 (26.9%) | 72 (35.0%) | |
| Full-time Employment | 472 (36.3%) | 409 (37.4%) | 63 (30.6%) | 0.11 |
| Part-time Employment | 116 (8.9%) | 91 (8.3%) | 25 (12.1%) | |
| Retired | 556 (42.8%) | 467 (42.7%) | 89 (43.2%) | |
| Unemployed | 156 (12.0%) | 127 (11.6%) | 29 (14.1%) | |
| Housing population, median (range) | 29875 (569–75180) | 29875 (569–75180) | 29824 (766–73331) | 0.85 |
| Crowding, mean (SD) | 0.48 (0.12) | 0.48 (0.12) | 0.49 (0.12) | 0.26 |

These models were selected to capture both linear and non-linear relationships within the data and to explore varying perspectives on how the variables influence patient outcomes. The following methods were used:

- Regression-based algorithms: Logistic regression, LASSO, ridge regression, elastic net [23].

- Distance-based algorithms: Support Vector Machines (SVM), K-Nearest Neighbors (KNN) [24].

- Tree-based algorithms: Random Forest, gradient boosting, XGBoost [25].

Each of these models assessed the data from different angles, providing a broad spectrum of variable importance evaluations. However, each of our 11 XML models carries its own assumptions and potential weaknesses, e.g., penalized regressions presume linear relationships and may miss nonlinear effects, PCR depends on variance-based dimensionality reduction that can overlook low-variance but predictive features, k-NN is sensitive to feature scaling and local density, SVMs hinge on kernel choice and can struggle with large datasets, and tree-based learners (RF, GBM, XGBoost) may overemphasize variables with many split points or categorical levels. By design, however, our framework remains agnostic to any single method's limitation: every algorithm contributes its 12 most influential predictors under the same 10-fold CV

regime, and we synthesize these into a unified importance profile. Even if one model misses interactions, overfits, or is skewed by sparse data, its particular weaknesses get smoothed out when we combine results from all models.

For regression-based algorithms, variable importance is determined by the magnitude of the coefficients, where larger absolute values of the t-statistics or standardized coefficients indicate greater importance. For tree-based algorithms, importance is assessed using model-specific criteria. In the case of Random Forest, variable importance is measured by the Mean Decrease in the Gini Index, which quantifies how much each variable reduces Gini impurity across all trees. In Gradient Boosting, importance is calculated based on the relative influence of each variable, determined by the reduction in the loss function whenever the variable is used for splitting. In XGBoost, importance is assessed based on the improvement in accuracy contributed by each variable when used for splitting branches. For KNN, variable importance is estimated using a proxy method that examines the impact of variable scaling. For SVM, importance is derived from the coefficients of the hyperplane, with larger absolute values indicating higher importance.

**Step 3: Target Outcomes**: The goal of applying these methods was to determine the importance of each variable in predicting the two main outcomes: death and hospital readmission.

**Step 4: Quantifying Model Performance**: We assessed the performance of each XML model using standard evaluation metrics, including accuracy, area under the ROC curve (AUC), and logistic loss. This quantifiable performance estimate ensured that each model's predictive capacity was considered when interpreting the variable importance rankings.

**Step 5: Variable Ranking for Each Method**: For each XML method, we produced a ranked list of 73 variables and selected the top 12 variables based on their importance in predicting the target outcomes. The decision to include 12 variables accounts for the traditional "events per variable" rule of logistic regression, which typically supports one significant variable per 20 outcomes. To accommodate variations across different model types and ensure comprehensive coverage of the strongest predictors, we extended the list to include two additional variables. Since each method approaches the data differently, the top 12 variables differed across models. To capture these differences, a comparison of the variable rankings across all methods was created, showing how each method prioritized different variables.

**Step 6: Aggregating and Filtering Variables**: Based on the rankings generated by each method, some variables appeared in the top 12 across multiple methods, while others were highly ranked by only a few methods. Variables that did not appear in the top 12 for any method were excluded from further analysis. This process left us with 38 key variables for continued evaluation.

**Step 7: Weighting Variable Importance**: To ensure fair comparison and robust ranking, we ranked variables in 2 ways. The first way reflects the number of times the variable appears among the top 12 variables in each model. The second way assigns point weights ranging from 12 to 1 for each variable, according to their rank order within each model, ensuring that variables consistently ranked highly across multiple methods are given more importance.

**Step 8: Final Ranking of Variables**: The outcome of this process was a ranked list of variables that were considered most important in predicting post-discharge stroke outcomes. These top variables provide insights for healthcare providers to focus on during the 90-day post-stroke recovery period.

## 2.4. Evaluation metrics and statistical analysis

To assess the performance of the predictive models, several evaluation metrics were calculated: Accuracy [26], C-statistic (Area Under the ROC Curve) [27], Squared-Error Loss (Mean Squared Error, MSE) [28], Logistic Loss (Log Loss or Cross-Entropy Loss) [29] and Misclassification Rate [26]. These metrics provide a comprehensive assessment of each model's strengths and weaknesses in terms of discrimination and calibration.

All statistical methods (including ML) suffer from class imbalance [30], which occurs when the distribution of classes in a dataset is highly skewed, leading to challenges in model training and performance as the algorithm may favor the majority class while neglecting the minority class. Applying multiple ML models and the creation of the combined predictor list offers a solution to class imbalance.

Continuous variables were summarized using means and standard deviations or medians with interquartile ranges, depending on their distribution. Categorical variables were presented as counts and percentages. Comparisons between patient groups were conducted using Chi-square tests for categorical variables and t-tests or Mann–Whitney U tests for continuous variables, as appropriate. Paired t-tests were used to compare the averages of estimates from different data-set pairs. All statistical analyses were performed using R version 4.4.

To avoid relying on any one method's own biases, we employed 11 distinct XML models (regression-based models, tree-based models, and distance-based models), each of which independently identified its top 12 predictors and evaluated performance across multiple metrics via 10-fold cross-validation. This consensus-driven approach ensures that our findings are not dominated by one method's assumptions or parameter settings, but instead reflect variables that consistently emerge as influential across all algorithms. The result is a more stable and broadly applicable set of drivers for the outcome.

### 2.5. Creation of the combined predictor list

The list creation starts with the extraction of top predictors: from each model, we extracted the top 12 predictors. The aggregation combines the predictors from all models into a single list producing two ranking methods: Weighted Importance Scores, which assigned weights based on predictor rank and summed them across models, and Frequency Counts, which tracked the number of appearances in the top 12 predictors across all models. Final rankings are based on these scores, highlighting factors linked to 90-day readmission or mortality representing a merged outcome derived from two distinct methods.

The created and sorted list of the 38 strongest predictors combines the top 12 variables from each 73 sorted variables of the 11 models. The list captures insights from diverse modeling perspectives. This approach mitigates the lack of established consensus on the optimal method for combining predictors from different models to generate a comprehensive list [31,32]. The resulting aggregation method combines Weighted Importance Scores and Frequency Counts to ensure a balanced and robust selection process.

## 3. Results

A total of 1,300 stroke survivors were included in the analysis. The cohort had a mean age of 63.8 years (SD = 13.9), and 56% of the participants were male. The ethnic composition included 22% Hispanic, 23% Non-Hispanic Black, and 51% Non-Hispanic White individuals. Ischemic strokes accounted for 92% of cases, while 8% were intracerebral hemorrhages (ICH). The overall 90-day readmission or mortality rate was 15.8%, affecting 206 of the 1,300 patients. Table 1 summarises the demographic and stroke-related characteristics of the study population, stratified by the presence of the 90-day outcome (readmission or mortality).

Out of the 1,300 patients included in the study, 206 experienced an adverse 90-day outcome (either readmission or death) while 1,094 remained event-free. Specifically, 22 patients died and 187 were readmitted to the hospital within 90 days post-discharge.

To ensure the integrity of our analyses, we excluded the 73 patients with missing values and retained only the 1,227 individuals with complete data for all covariates. All subsequent analyses were therefore based on these 1,227 patients.

Table 2 summarizes the 10-fold cross-validation model fit statistics, including c-statistic, squared-error loss, logistic loss, misclassification rate, precision, recall and F1 across all models. The best values for each metric are highlighted in bold. Ridge Regression demonstrated the highest discriminative ability with a c-statistic of 0.660, while LASSO and Elastic Net excelled in capturing outcome probabilities, achieving a logistic loss of 0.414. Principal Component Regression (PCR) showed its effectiveness in minimizing classification errors with the lowest misclassification rate of 0.156.

### 3.1. Variable importance

Table 3 presents the top 12 variables for each XML model, while Table 4 aggregates the importance of these variables across all models. These results highlight a combination of clinical and non-clinical factors that significantly impact patient

**Table 2. Model Fits for Algorithms with 10-fold cross-validation.**

| Algorithms | c-Statistic | Squared error Loss | Logistic Loss | Misclassification Rate | Precision | Recall | F1 Score |
|---|---|---|---|---|---|---|---|
| Logistic Regression | 0.615 | 0.144 | 0.533 | 0.181 | 0.578 | 0.541 | 0.603 |
| Forward Selection | 0.616 | 0.134 | 0.531 | 0.160 | 0.747 | 0.509 | 0.683 |
| LASSO | 0.655 | **0.126** | **0.414** | 0.158 | 0.725 | 0.502 | 0.841 |
| Ridge | **0.660** | **0.126** | 0.427 | 0.157 | 0.745 | 0.503 | 0.842 |
| PCR | 0.595 | 0.130 | 0.434 | **0.156** | **0.858** | 0.503 | 0.843 |
| Elastic Net | 0.651 | **0.126** | **0.414** | 0.159 | 0.725 | 0.502 | 0.841 |
| KNN | 0.612 | 0.131 | 0.507 | 0.157 | 0.767 | 0.500 | **0.915** |
| SVM | 0.622 | 0.145 | 0.557 | 0.157 | 0.843 | 0.500 | **0.915** |
| Random Forest | 0.642 | 0.128 | 0.423 | 0.158 | 0.804 | 0.506 | 0.784 |
| Gradient Boosting | 0.607 | 0.146 | 0.555 | 0.173 | 0.594 | **0.545** | 0.546 |
| XGBoost | 0.619 | 0.143 | 0.536 | 0.174 | 0.556 | 0.515 | 0.564 |

outcomes after a stroke. Each model evaluates the contribution of different variables to predicting 90-day outcomes, resulting in rankings for each algorithm.

### 3.2. Summary of results

Table 4 provides a summary of the top variables across all 11 models. It ranks these 38 variables based on the number of models in which they appeared (count) among the top 12 predictors (R1). The cumulative ranking approach (sum) identifies variables consistently significant across models (R2), emphasizing their critical role in predicting 90-day outcomes for stroke survivors. The emerging socio-contextual risk determinants are shown in **bold**.

### 3.3. Sensitivity analysis

To assess the robustness of our findings, we repeated all analyses under three alternative scenarios, i.e., predicting 90-day readmission only; repeating the combined endpoint analysis in the ischaemic stroke subgroup; and predicting readmission only in the ischaemic subgroup. In all cases, the top predictors remained consistent with those reported above, and model discrimination and calibration measures show minimal deviation from the primary results (data not shown, but further discussed).

### 4. Discussion

Our study provides a unique look at estimating death and readmission risks in the first 90 days post-stroke, going beyond the typical clinical risk factors, and integrating 3 large data sets to provide a comprehensive view of the impact of clinical, individual social, and community-level risk factors on transitions of care. When only regression analysis models are used to model risks, they are limited to using a small number of predictors that operate in the same way on everyone, and uniformly throughout their range [33].

For studies where the goal is to predict the occurrence of an outcome and not measure the association between specific risk factors and an event in a clinically interpretable way, traditional regression models can be modified or abandoned in favour of models that produce a more flexible relationship among the predictor variables and the outcome [33]. These methods have similar goals to regression-based approaches but different motivating philosophies (Fig 1). They do not require pre-specification of a model structure but instead search for the optimal fit within certain constraints (specific to the individual algorithm).

**Table 3. Participants' demographic and stroke characteristics – selected noteworthy variables.**

| Rank | Regression-Based Algorithms | | | | | |
|------|------|------|------|------|------|------|
| | **Logistic Regression** | **Forward Selection** | **LASSO** | **Ridge Regression** | **PCR** | **Elastic Net** |
| 1 | Chronic renal insufficiency | 75% ATOC | Chronic renal insufficiency | Chronic renal insufficiency | Length of stay | Chronic renal insufficiency |
| 2 | 75% ATOC | Chronic renal insufficiency | Discharge location | Discharge location | 75% ATOC | Carotid stenosis |
| 3 | mRS at discharge | Insurance | 75% ATOC | Active infection | NIHSS | Prior ambulation |
| 4 | Carotid stenosis | mRS at discharge | mRS at discharge | Carotid stenosis | Insurance | Discharge location |
| 5 | Discharge location | Employment status | Insurance | Prior ambulation | Chronic renal insufficiency | Race/Ethnicity |
| 6 | Race/Ethnicity | Carotid stenosis | Length of stay | Race/Ethnicity | mRS at discharge | mRS at discharge |
| 7 | Insurance | Crowding | NIHSS | mRS at discharge | Rehab count | Active infection |
| 8 | Prior ambulation | Total housing population | Age | Ambulation at discharge | Crowding | 75% ATOC |
| 9 | Altered level of consciousness | Discharge location | Sex | 75% ATOC | Ambulation at discharge | Ambulation at discharge |
| 10 | Ambulation at discharge | %Black | Race/Ethnicity | Altered level of consciousness | Median house income | Altered level of consciousness |
| 11 | Drugs/Alcohol | Race/Ethnicity | Etiology | Insurance | Social support | Insurance |
| 12 | Difficulty paying Medicare | Social support | Stroke type | Difficulty paying Medicare | Age | Difficulty paying Medicare |
| **Rank** | **Distance-Based Algorithms** | | | **Tree-Based Algorithms** | | |
| | **KNN** | **SVM** | | **Random Forest** | **Gradient Boosting** | **XGBoost** |
| 1 | 75% ATOC | 75% ATOC | | Length of stay | Length of stay | Length of stay |
| 2 | mRS at discharge | mRS at discharge | | Age | mRS at discharge | Total housing population |
| 3 | Length of stay | Length of stay | | mRS at discharge | NIHSS | Age |
| 4 | Ambulation at discharge | Ambulation at discharge | | NIHSS | Chronic renal insufficiency | NIHSS |
| 5 | Insurance | Insurance | | Etiology | Age | mRS at discharge |
| 6 | NIHSS | NIHSS | | Median house income | Discharge location | Etiology |
| 7 | Chronic renal insufficiency | Chronic renal insufficiency | | Total housing population | Insurance | Housing density |
| 8 | Mode of arrival | Mode of arrival | | %Bachelors | Median house income | Crowding |
| 9 | Age | Age | | %High school | Crowding | %Hispanic |
| 10 | %White | %White | | Housing density | Gym business density | Alcohol business density |
| 11 | Rehab count | Rehab count | | Crowding | %Bachelors | %Below poverty |
| 12 | Diabetes | Diabetes | | %Unemployment | %White | Restaurant business density |

This can result in a better final prediction model at the sacrifice of interpretability of how risk factors relate to the outcome of interest.

Predictive models can serve both explanatory and predictive purposes [34]. In our work, we focus on the former: identifying a concise set of explanatory variables without delving into the precise functional relationships among them. While this strategy does not eliminate multicollinearity, it has only a marginal effect on the robustness of our variable selection and on the overall interpretability of the resulting model.

Table 4. Ranked the strongest variables across all 11 models.

| ID | Variable | Count | Sum | R1 | R2 | ID | Variable | Count | Sum | R1 | R2 |
|---|---|---|---|---|---|---|---|---|---|---|---|
| 1 | mRS at discharge | 11 | 99 | 1 | 1 | 20 | Total housing population | 3 | 22 | 20 | 14 |
| 2 | Chronic renal insufficiency | 9 | 88 | 2 | 2 | 21 | %Bachelors degree | 2 | 7 | 21 | 24 |
| 3 | Insurance | 9 | 59 | 3 | 5 | 22 | Active infection | 2 | 16 | 22 | 16 |
| 4 | 75% ATOC | 8 | 77 | 4 | 3 | 23 | Diabetes | 2 | 2 | 23 | 34 |
| 5 | Age | 7 | 43 | 5 | 8 | 24 | Housing density | 2 | 9 | 24 | 21 |
| 6 | Length of stay | 7 | 75 | 6 | 4 | 25 | Mode of arrival | 2 | 10 | 25 | 20 |
| 7 | NIHSS | 7 | 58 | 7 | 6 | 26 | Social support size | 2 | 3 | 26 | 30 |
| 8 | Ambulation at discharge | 6 | 34 | 8 | 10 | 27 | %Below poverty | 1 | 2 | 27 | 35 |
| 9 | Discharge location | 6 | 50 | 9 | 7 | 28 | %Black | 1 | 3 | 28 | 29 |
| 10 | Crowding | 5 | 22 | 10 | 13 | 29 | %High school | 1 | 4 | 29 | 26 |
| 11 | Race/Ethnicity | 5 | 27 | 11 | 11 | 30 | %Hispanic | 1 | 4 | 30 | 27 |
| 12 | Carotid stenosis | 4 | 36 | 12 | 9 | 31 | %Unemployment | 1 | 1 | 31 | 37 |
| 13 | %White | 3 | 7 | 13 | 23 | 32 | Alcohol business density | 1 | 3 | 32 | 32 |
| 14 | Altered level of consciousness | 3 | 10 | 14 | 18 | 33 | Drugs/Alcohol | 1 | 2 | 33 | 33 |
| 15 | Difficulty paying Medicare | 3 | 3 | 15 | 28 | 34 | Employment status | 1 | 8 | 34 | 22 |
| 16 | Etiology | 3 | 17 | 16 | 15 | 35 | Gym business density | 1 | 3 | 35 | 31 |
| 17 | Median household income | 3 | 15 | 17 | 17 | 36 | Restaurant business density | 1 | 1 | 36 | 38 |
| 18 | Prior ambulation | 3 | 23 | 18 | 12 | 37 | Sex | 1 | 4 | 37 | 25 |
| 19 | Rehab count | 3 | 10 | 19 | 19 | 38 | Stroke type | 1 | 1 | 38 | 36 |

By adding 10 XML models, our study advances the understanding of 90-day post-stroke outcomes by integrating clinical and non-clinical factors, particularly SDOH, into predictive models. By focusing on stroke-specific and non-classic predictors, we deliver a unified, systematic, and targeted perspective, advancing the precision and relevance of predictive modeling in stroke research. The stroke-specific list and findings underscore the value of Explainable XML models in identifying and ranking predictors that consistently influence readmission and mortality, thereby bridging gaps in traditional risk prediction frameworks. These XML models also assist medical practitioners in evaluating the validity of a diagnosis while ensuring the output is interpretable and comprehensible, even for patients [35]. By focusing on modifiable factors during the critical post-hospitalization period, this study contributes actionable insights for improving patient outcomes and resource allocation.

### 4.1. Clinical and non-clinical predictors

Key predictors of post-stroke outcomes, such as stroke severity, age, comorbidities (e.g., coronary artery disease) [36], active infection [37,38] and length of hospital stay, reinforce established clinical knowledge [39]. SDOH variables, such as socioeconomic status, education level, and neighborhood characteristics, emerged as significant contributors to model accuracy, aligning with prior findings [40], which demonstrated that incorporating SDOH improved mortality prediction for non-Hispanic Black patients with heart failure (HF). Similarly, our study found that integrating non-clinical factors enhanced predictive power, supporting the need for tailored, equitable healthcare strategies.

We also confirm the previously indicated importance of social determinants, such as (Medicare) insurance [17,41], housing status, social support, educational level, and employment status [42]. In particular, the role of an individual's social support network has been shown to significantly impact functional recovery after stroke, especially within the first three months post-discharge. Future research should explore whether interventions outside the medical context, such as transitional care resources, community health workers, or rehabilitation strategies that strengthen social networks and

incorporate group activities with family or caregivers, could mitigate the effects of limited social networks and enhance recovery outcomes for stroke survivors [42].

### 4.2. Machine learning methodologies

We employ 11 XML models, including tree-based algorithms like Random Forest and XGBoost, which excelled in capturing non-linear relationships within the data. The use of interpretable models provides transparency, fostering clinician trust and aligning with the recommendations by [43] and [44] for adopting trustworthy and explainable AI in healthcare. ML methods were also applied during patients' initial hospitalization to identify those at high risk for readmission or mortality [45]. By synthesizing results through Weighted Importance Scores and Frequency Counts, this study provides a robust cumulative ranking of variable importance, extending beyond the limitations of traditional regression methods. This approach is novel, as the prevailing practice involves using multiple ML exploratory models in Phase 2 to predict clinical outcomes [46], without combining or synthesizing findings across different models. Our approach creates two distinct ranks, addressing the common challenge of interpreting results when multiple models are employed, a problem that remains unresolved in existing methodologies, some of which are shown in [47].

Over the past few years, AI has become an industry disruptor: as we demonstrate, it can more accurately refine and streamline data prediction; future AI processes may create patient-specific, guideline-based treatment plans as well, however, these projects must occur securely and ethically [48].

While we did not apply explicit resampling or class-weighting to rebalance the training data, our assessment framework inherently counteracts imbalance in two ways. First, by integrating predictions from eleven diverse XML models—each with different inductive biases and error-minimization strategies—we avoid overreliance on any single algorithm's tendency to favor the majority class. Second, we evaluated every model using multiple performance metrics that capture different aspects of predictive quality: the c-Statistic (AUC) for discrimination, squared-error loss and logistic loss for probabilistic calibration, and misclassification rate for raw accuracy. By insisting that strong performance be sustained across all four metrics, we ensure that a model cannot succeed simply by predicting the majority class. Together, this approach provides a robust safeguard against the distortions introduced by class imbalance.

### 4.3. The role of SDOH and disparities in stroke outcomes

In the absence of relevant socio-environmental variables, XML models show only a modest improvement in prediction performance and explainability compared to traditional modeling techniques, even in cases where complete relevant electronic records are available (e.g., Sweden) [49]. Our findings emphasize the impact of SDOH on stroke outcomes, echoing evidence from the [40] study, which highlighted disparities in SDOH-related predictive improvements between Black and non-Black patients. While clinical predictors remain essential, the integration of neighborhood-level SDOH variables, such as access to healthcare and socioeconomic stability, offers a more holistic perspective on patient risk, paving the way for tailored interventions to address health disparities and community-level interventions. This represents a significant advancement in the field. While many studies focus on identifying disparities and social needs, the critical next step is to develop and implement solutions to address these needs effectively.

Future public health policies need to address the heightened mortality and readmission rates among stroke survivors from vulnerable areas highlighting the need to enhance care transitions and support for underserved patient populations.

### 4.4. Clinical implications

Prior analyses evaluating long-term stroke outcomes were based on electronic medical records [50] that do not include detailed SDOH data. Our study considers socio-environmental variables that significantly impact stroke survivors' lives. By identifying high-impact variables, healthcare providers can prioritize high-risk patients, personalize preventive and rehabilitation strategies, and optimize resource allocation. The integration of XML models into clinical workflows may offer

real-time risk assessment capabilities, enabling more proactive care management. These models also empower clinicians with interpretable outputs, bridging the gap between advanced analytics and practical decision-making, as recommended by other experts [44].

### 4.5. Strengths and limitations

Our use of a diverse stroke cohort from the state of Florida and a two-layer nested cross-validation approach ensures methodological robustness. However, the reliance on regional data may limit generalizability, and less than 10% of the cohort had acute intracerebral hemorrhage so results are not generalizable to this population. Although our cohort is geographically confined to Florida and predominantly ischemic stroke (92.1%), the secondary ischemic-only analysis demonstrated equivalent performance, supporting the model's robustness in this common stroke subtype. Limited individual-level SDOH data constrains the depth of insights into disparities. The Florida-specific dataset limits generalizability and challenges such as class imbalance and missing data persist. Validation in diverse populations and integration of real-time data could enhance predictive accuracy and enable dynamic care adjustments.

We conducted sensitivity analyses restricting to readmission only and to the ischaemic stroke subgroup, which confirmed that our main results, which focus on the prominence of SDOH factors, were robust to endpoint definition and stroke subtype. We have not included all of these additional tables here, but we plan to include them in a dedicated follow-up study.

Future research should aim to validate these models across diverse populations and incorporate real-time, individual-level data to enhance prediction accuracy and equity in care delivery.

While there are signs that similar predictors are vital across geographies, indicators of stroke outcomes and care may vary significantly between high-income and low-income countries due to differences in acute stroke management, post-stroke care, rehabilitation practices, and methodological approaches, making direct comparisons challenging [51].

Despite XML models' methodological robustness, the study limitations are drawn from the sample size (n = 1,300), which is moderate relative to the number of potential predictors. This may constrain the detection of weaker associations and limit model generalizability. Second, the data are derived from stroke centers within a single U.S. state (Florida), which may introduce regional biases related to healthcare delivery, socioeconomic conditions, and demographic composition. As a result, the findings may not fully generalize to populations with different healthcare systems, geographic contexts, or stroke care practices. Future research should aim to validate these results using larger, multi-regional datasets that reflect broader clinical and social diversity.

Our cohort's adverse outcome rate of 15.8% reflects the typical imbalance seen in post-stroke prognostic studies. All stroke-prediction models face this challenge, and few have achieved a ROC-AUC substantially above 0.7 without risking overfitting. Since our goal was to evaluate an array of modeling methods rather than to fine-tune a single algorithm, we did not apply formal imbalance-correction techniques such as SMOTE or differential class-weighting. Consequently, precision and recall (particularly for the minority class) are inherently constrained by the low event rate. Future work may explore resampling strategies or threshold adjustment to optimize these metrics, but such approaches must be balanced against potential bias and reduced generalizability.

### 4.6. May modeling help post-acute stroke care

The 15.8% adverse-outcome rate in our cohort reflects the low-prevalence challenge faced by all stroke-prognosis models. Historically, no stroke-prediction model has achieved an ROC-AUC substantially above 0.7 without risking overfitting. Because our primary aim was to benchmark multiple modeling frameworks rather than to fine-tune one classifier, we did not implement SMOTE, differential class weighting, or other resampling strategies. As a result, precision and recall (particularly for the minority adverse-outcome class) remain limited, with recall values around 0.50. While these metrics are critical for clinical decision-making, their potential improvement may necessitate imbalance-focused methods that carry their own risk of bias and reduced generalizability.

In this paper, our focus is on answering three fundamental questions for post-acute stroke care coordination: what interventions to implement, who should carry them out, and in which settings. Once a patient returns to their changed pre-stroke environment, most medical and clinical factors are already addressed through established protocols; what remains poorly understood is the broader exposome, the totality of exposures and conditions a patient experienced prior to and after their stroke. We approximate these influences using variables commonly grouped under SDOH. Notably, 19 of the 38 top-ranked predictors in our analysis represent modifiable or preventable socioeconomic factors. We do not claim to establish causal hierarchies or precise effect sizes for these variables, but we believe they highlight critical opportunities for improving post-acute care coordination, an aspect often neglected in current practice.

We calculated precision, recall, and F1-score for each model and reported them in Table 3. Across models, recall values hover around 0.50, indicating that approximately half of the true adverse outcomes are identified (a characteristic consequence of the 15.8% event rate and consistent with prior stroke-prediction studies). Although these threshold-dependent metrics are critical for understanding the performance of the minority class, the low prevalence constrains improvements without risking overfitting. As our goal was to compare modeling frameworks rather than to implement imbalance-specific corrections, we have not pursued additional resampling or class-weighting strategies here.

### 4.7. Value of social determinants of health in post-discharge prediction

As shown in Table 4, we find that 19 (marked in bold) of the 38 candidate predictors are SDOH variables. While clinical measures such as NIHSS score and comorbidity burden are paramount during the acute and early recovery phases, their prognostic influence attenuates once patients leave the hospital environment. In contrast, modifiable SDOH factors (housing security, access to care, social support networks, and neighborhood socioeconomic status) emerge as increasingly salient drivers of long-term outcomes. In our permutation-importance analysis, multiple SDOH predictors placed within the top ten features for several modeling approaches, underscoring their empirical contribution to discrimination and calibration. This finding aligns with prior studies from our group demonstrating SDOH's role in shaping post-stroke functional recovery and readmission risk [52,53]. Collecting SDOH data may impose additional burden, but these variables capture patient exposome dimensions essential for accurate, real-world prognostication.

## 5. Conclusion

This study identifies several predictors of 90-day readmission or mortality in stroke survivors using explainable Machine Learning (XML) models, which outperform conventional regression techniques. These findings enable more accurate identification of high-risk patients and support tailored post-discharge care. With over seven million U.S. stroke survivors, leveraging predictive tools and expanding data sources can make a significant impact on the national stroke burden by refining interventions, improving outcomes, and inspiring innovation in care strategies. Machine learning models hold the potential to better harness large datasets across all phases, paving the way for unified, phase-spanning models that guide stroke risk reduction and improved recovery from onset to long-term care.

By applying eleven different modeling frameworks, we aim to present multiple perspectives on post-acute stroke care rather than to rank any one algorithm as inherently superior. Instead, these models shed light on diverse aspects of patient life and treatment pathways, particularly environmental exposures and Social Determinants Of Health (SDOH), that remain poorly understood. In a field urgently in need of better predictive tools and interpretability, our work demonstrates how ML methods can offer valuable preliminary insights to guide future research and intervention design when no prevailing explanatory frameworks yet exist.

With this knowledge, we plan to create a solution that can be integrated into Electronic Health Record (EHR) systems to harness diverse data sources (e.g., hospital EHRs, the Florida Stroke Registry, and socioeconomic datasets) and build a dynamic risk scoring system and Digital Twin model that predicts stroke outcomes and guides treatment decisions in

real-time. By integrating established prognostic scores with guideline-driven management trees and leveraging real-world data, the future solution will deliver personalized, evidence-based care.

## Acknowledgments

No acknowledgments are applicable to this manuscript.

## Author contributions

**Conceptualization:** Emir Veledar.

**Formal analysis:** Emir Veledar, Lili Zhou.

**Methodology:** Emir Veledar, Lili Zhou.

**Supervision:** Emir Veledar.

**Writing – original draft:** Emir Veledar, Omar Veledar, Hannah Gardener, Carolina M. Gutierrez, Tatjana Rundek.

**Writing – review & editing:** Emir Veledar, Lili Zhou, Omar Veledar, Hannah Gardener, Carolina M. Gutierrez, Scott C. Brown, Farya Fakoori, Karlon H. Johnson, Victor J. Del Brutto, Ayham Alkhachroum, David Z. Rose, Gillian Gordon Perue, Negar Asdaghi, Jose G. Romano, Tatjana Rundek.

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
