## [Decision Letter · Decision Letter 0]

13 May 2025

Dear Dr. Veledar,

Thank you for submitting your manuscript to PLOS ONE. After careful consideration, we feel that it has merit but does not fully meet PLOS ONE’s publication criteria as it currently stands. Therefore, we invite you to submit a revised version of the manuscript that addresses the points raised during the review process.

The reviewers found the study important and methodologically strong but identified key issues that must be addressed for acceptance. Required changes include clarifying the outcome definition, detailing data preprocessing, addressing class imbalance with appropriate metrics, and adding supplemental materials defining predictors and reporting full model outputs. The inclusion of hemorrhagic strokes should be justified or evaluated through sensitivity analysis. While the use of SDOH is a strength, the discussion should acknowledge their limited impact on model performance and clarify their value.

We look forward to receiving your revised manuscript.

Kind regards,

Noah Hammarlund

Academic Editor

PLOS ONE

Journal Requirements:

3. For studies involving third-party data, we encourage authors to share any data specific to their analyses that they can legally distribute. PLOS recognizes, however, that authors may be using third-party data they do not have the rights to share. When third-party data cannot be publicly shared, authors must provide all information necessary for interested researchers to apply to gain access to the data. (https://journals.plos.org/plosone/s/data-availability#loc-acceptable-data-access-restrictions) 

Reviewers' comments:

Reviewer's Responses to Questions

**Comments to the Author**

1. Is the manuscript technically sound, and do the data support the conclusions?

Reviewer #1: Yes

Reviewer #2: Partly

2. Has the statistical analysis been performed appropriately and rigorously?

Reviewer #1: Yes

Reviewer #2: No

3. Have the authors made all data underlying the findings in their manuscript fully available?

Reviewer #1: Yes

Reviewer #2: Yes

4. Is the manuscript presented in an intelligible fashion and written in standard English?

Reviewer #1: Yes

Reviewer #2: Yes

Reviewer #1: See attachment for additional information. Three areas are outline that should be addressed by the authors before the publication criteria are fully satisfied.

This study uses a rigorous analytic approach of multiple XML models, including regression-based, tree-based, and distance-based algorithms. This diversity in modeling approaches enhances the robustness of the findings. Using 10-fold cross-validation ensures that the models are tested on different subsets of the data, reducing the risk of overfitting. The study uses a combined ranking approach, incorporating Weighted Importance Scores and Frequency Counts to identify significant predictors across models. The performance of each model is assessed using standard evaluation metrics, including accuracy, area under the ROC curve (AUC), squared-error loss, logistic loss, and misclassification rate.

However, I strongly encourage the inclusion of additional information in the study to increase the interpretability and understanding of the findings. By incorporating additional information, robustness tests, and replicability measures, the study can further enhance its credibility and impact in the field of stroke research.

1. Detailed Description of Data Preprocessing:

• The manuscript should include a more detailed description of the data preprocessing steps, such as handling missing values, normalization, and applying feature engineering techniques.

2. Model Hyperparameters:

• Providing information on the hyperparameters used for each XML model would enhance the study's transparency and replicability.

3. Sensitivity Analysis:

• Including a sensitivity analysis to show how changes in model parameters or the inclusion/exclusion of certain variables affect the results would strengthen the robustness of the findings.

I would also encourage the authors to discuss the weaknesses and limitations of their modeling approach and their data sources. The following areas should be addresses:

1. Sample Size and Generalizability:

• Acknowledge the limitations related to sample size and the regional focus of the study. Discuss how these factors might affect the generalizability of the findings.

2. Class Imbalance:

• Address the issue of class imbalance in the dataset and how it was managed. Discuss any potential impacts on model performance and the interpretation of results.

3. Model Limitations:

• Highlight any limitations of the XML models used, such as their reliance on specific types of data or potential biases in variable selection.

4. Data Availability:

• Mention any restrictions on data availability and how they might limit the ability of other researchers to replicate the study.

Finally, the only thing missing from this study is a complete outline of the potential implications of these findings. I would like to know how these conclusions can be used without the following areas:

1. Clinical Practice:

• Discuss how the findings can be applied in clinical practice to improve post-stroke care, resource allocation, and patient outcomes. Highlight any specific recommendations for healthcare providers.

2. Policy and Public Health:

• Explore the implications for public health policy, particularly in addressing socioeconomic disparities and improving care transitions for stroke survivors.

3. Future Research:

• Identify areas for future research, such as developing interventions based on the identified predictors, validating the models in different populations, and exploring additional variables.

By including these additional details, the manuscript can provide a more comprehensive and nuanced understanding of the study's findings, their implications, and the context in which they were derived. This will also help address potential concerns and enhance the research's replicability and applicability.

Reviewer #2: This is a well written, important study that attempts to identify important clinical and non-clinical risk factors that help predict mortality and rehospitalization for patients post-stroke using a consolidated approach of multiple explainable ML techniques. I think the research has merit but there are a number of statistical problems that should be addressed before a satisfactory evaluation of this manuscript.

1. The introduction is well written and clear, there are a number of statements that would benefit from citations (e.g. while recent studies have not substantially improved prediction intervals...), Furthermore, the introduction could present more substantial information regarding performance of classifiers from previous studies as well as problems and limitations that makes this study unique (such as the inclusion of SDOH variables into these types of models)

2. It is unclear how the outcomes were combined into a single variable, what is the frequency for each outcome in the cohort?

3. Class Imbalance: The manuscript suggests that applying multiple machine learning (ML) methods and combining predictors helps address class imbalance. However, I am not convinced this alone mitigates the issue, especially in the absence of intrinsic strategies such as resampling, class weighting, or appropriate performance metrics. Furthermore, this assertion lacks citations, which weakens the credibility of the claim. I recommend supporting this statement with relevant literature or clarifying the methodological justification.

4. I would strongly recommend restricting the analysis to ischemic strokes. The dataset contains only ~100 cases of intracerebral hemorrhage, which likely have distinct clinical and demographic predictors compared to ischemic strokes. Including them may introduce noise that could impair the model's discriminatory performance and interpretability.

5. Table 1 Statistical Analysis: The manuscript does not clearly indicate what statistical tests were applied to generate Table 1. Given the presence of varied data types (e.g., means, medians with ranges, and percentages), it is important to specify whether the appropriate tests were used for each variable type. Additionally, it is unclear if corrections for multiple comparisons were performed.

6. Lasso, ridge and elastic nets are not distinct types of ML classifiers from logistic regression, rather there are regularization techniques aimed to reduce the important of coefficients by either shrinking their values or removing them altogether. Including all four variants (standard logistic regression, LASSO, ridge, and elastic net) in an ensemble may introduce redundancy, as they share the same underlying structure and linear decision boundaries. While the ensemble includes other diverse model types such as SVM, KNN, and decision tree-based methods, it may be more effective to select one or two representative logistic regression variants to avoid overemphasizing a single modeling framework and to maintain better balance across model types.

7. ML Model Performance: It would be valuable to include a supplemental table showing the full results for each machine learning model (Coefficients, predicted probabilities, odds ratios, etc). Was there an evaluation of the performance using the selected either 12 or 38 variables? is there an improvement in the prediction when using the most important variables?

8. Imbalanced Data: The dataset is highly imbalanced, yet the performance metrics presented do not adequately reflect this. Metrics such as precision, recall, and F1-score—particularly for the minority class—are essential in this context. Including confusion matrices would also help readers assess the classifiers' performance across both classes.

9. On Multicollinearity: The issue of multicollinearity among predictors is not addressed. Given the number of variables and the potential for correlated features, I recommend performing and reporting multicollinearity diagnostics (e.g., variance inflation factors), particularly for models like logistic regression where interpretation of coefficients is important.

10. On Clarity and Definition of Predictors:The manuscript lacks a clear definition and description of the predictors used. Many predictor acronyms are not defined in the text, and categorical variables are not described in terms of their levels or categories. I suggest including a supplemental table that clearly lists all predictors, defines each acronym, and indicates the data type (continuous, categorical), as well as levels for categorical variables.

11. Unfortunately, this study did not improve upon the performance metrics reported in previous analyses, despite increasing the number of variables by incorporating several non-clinical factors, such as social determinants of health (SDOH). Since the addition of these variables did not enhance model performance, it raises the question of their practical value—particularly given that SDOH data are often more difficult to obtain than standard clinical variables.

12. There are a few grammatical errors: last word of introduction unballanced and in the sentence... The TCSD-S enrollee data from 10 collaborating comprehensive stroke canters (is it centers)

**Do you want your identity to be public for this peer review?** For information about this choice, including consent withdrawal, please see our Privacy Policy

Reviewer #1: No

Reviewer #2: No

---

## [Author Response · Author response to Decision Letter 1]

9 Jun 2025

All our comments are already included in attached answer to reviewers. We tried to answer all questions and suggestions one by one.

---

## [Decision Letter · Decision Letter 1]

17 Jun 2025

Dear Dr. Veledar,

Thank you for submitting your manuscript to PLOS ONE. After careful consideration, we feel that it has merit but does not fully meet PLOS ONE’s publication criteria as it currently stands. Therefore, we invite you to submit a revised version of the manuscript that addresses the points raised during the review process.

We look forward to receiving your revised manuscript.

Kind regards,

Noah Hammarlund

Academic Editor

PLOS ONE

Journal Requirements:

Additional Editor Comments:

Thank you for your revised manuscript. The study presents a thoughtful and innovative approach, and both reviewers noted major improvements. However, they also raise several remaining concerns that must be addressed before the manuscript can be considered for publication.

Where the reviewers raise methodological gaps (e.g., around class imbalance, model justification, or variable inclusion), you should either (a) conduct and report targeted additional analyses where feasible, or (b) clearly explain and justify your approach, and acknowledge resulting limitations where appropriate.

Please address the following points:

Composite Outcome: Justify the combination of readmission and mortality into a single outcome, especially given their differing frequencies. If separate models are not feasible, acknowledge this and discuss the implications as a limitation.

Stroke Type and Generalizability: Either conduct a sensitivity check restricted to ischemic stroke or justify your inclusion of ICH cases. Generalizability limitations due to geography and stroke subtype distribution should be made more explicit in the abstract and discussion.

Class Imbalance: The use of multiple models and evaluation metrics does not directly address class imbalance. Metrics like AUC and log loss can obscure performance issues on the minority class. Please provide a clearer discussion of the implications of imbalance and justify the decision not to apply standard techniques (e.g., weighting or resampling). Highlight how this may affect interpretation of metrics such as recall and variable rankings.

Multicollinearity: For regression-based models, address concerns about multicollinearity. Either assess and report relevant diagnostics (e.g., VIF, correlation checks), or justify your approach and acknowledge interpretability limitations.

Value of SDOH Variables: You make strong claims about the contribution of SDOH to model performance. While these variables appear in your ranked lists, the added predictive value is not directly assessed. We recommend a simple sensitivity analysis comparing model performance with and without SDOH predictors. If you do not include this, revise your language to more cautiously reflect what is shown, and note this limitation.

Explainability and Interpretation: Since the study emphasizes explainable ML, include a brief narrative interpretation of top predictors (e.g., mRS, ATOC) and how they could inform clinical decision-making.

Scope of Claims: Please revisit some of the manuscript’s stronger claims — such as statements that XML methods “enhance prediction” in small, imbalanced samples. Without comparative analyses or performance improvement tests, such claims should be tempered to reflect the more exploratory and descriptive nature of the work. Strengthening the limitations section will help appropriately frame the contribution.

We look forward to your resubmission.

Reviewers' comments:

Reviewer's Responses to Questions

**Comments to the Author**

Reviewer #1: (No Response)

Reviewer #2: (No Response)

2. Is the manuscript technically sound, and do the data support the conclusions?

Reviewer #1: Yes

Reviewer #2: Partly

3. Has the statistical analysis been performed appropriately and rigorously?

Reviewer #1: Yes

Reviewer #2: Yes

4. Have the authors made all data underlying the findings in their manuscript fully available?

Reviewer #1: Yes

Reviewer #2: Yes

5. Is the manuscript presented in an intelligible fashion and written in standard English?

Reviewer #1: Yes

Reviewer #2: Yes

Reviewer #1: This study uses explainable machine learning (XML) to identify clinical and non-clinical predictors of 90-day readmission and mortality among stroke survivors. Drawing on data from the Transitions of Care Stroke Disparities Study (TCSD-S), the authors analyzed outcomes for 1,300 patients using 11 machine learning models. The study integrates information from clinical registries, neighborhood-level social determinants of health (SDOH), and transition-of-care metrics. The key contribution is a composite list of 38 variables, derived from model consensus, that predict adverse post-stroke outcomes. The authors argue for the superiority of XML over traditional regression methods, highlighting the added value of SDOH in predictive modeling and the potential application in digital health systems.

Contribution of the Methodology, Insights, Conclusions:

1) The manuscript extends current clinical studies by integrating socioeconomic, neighborhood, and behavioral characteristics into the machine learning framework to enhance model relevance.

2) The authors clearly describe model selection, performance metrics, and variable importance aggregation (frequency count and weighted score), providing transparency and reproducibility.

3) The application of 11 XML models to identify consensus predictors is a novel and methodologically rigorous approach to multidimensional health outcome modeling that is strengthened by the 10-fold cross-validation and inclusion of multiple performance metrics (C-statistic, log loss, precision, etc.).

4) These findings could inform risk-stratification tools and targeted post-discharge interventions.

Additional Clarity Needed Before Publication

1) Outcome Construction: The composite outcome of “90-day readmission or mortality” is not adequately justified. The frequency of each outcome (readmission vs. death) is provided late in the paper (n=22 deaths, n=187 readmissions), suggesting imbalanced outcome events. Provide a more explicit rationale for combining these outcomes or present separate models for each. At minimum, sensitivity analyses using each outcome separately would enhance interpretability.

2) Additional Preprocessing Details: While the authors briefly mention normalization and the exclusion of 73 individuals due to missing data, there is insufficient detail on how missing values were handled or imputed. It would be beneficial if the authors explicitly described missing data patterns, imputation methods (if used), and any variable selection criteria. A flowchart could help illustrate data preprocessing.

3) Limited Generalizability: The sample is geographically confined to Florida, and 92% of strokes are ischemic. Only ~100 ICH cases are included, limiting broader applicability. Please consider performing a secondary analysis that is restricted to ischemic stroke only or provide stratified results. Discuss generalizability limitations more prominently in the abstract and conclusion.

4) Imbalance: The adverse outcome rate is relatively low (15.8%), yet the authors do not apply formal balancing techniques (e.g., SMOTE, class weights). It would be beneficial if the authors included an additional discussion on how class imbalance may or may not influence metrics such as precision and recall.

5) Explainability: The manuscript emphasizes “explainable” ML but does not attempt to interpret the most influential variables beyond summary tables. I suggest including a short narrative interpretation of how top predictors could inform clinical decision-making.

Reviewer #2: The manuscript has improved dramatically from the first submission, I believe it is a stronger work either analytically as well as conceptually. The changes to the text and analysis have improved the credibility to the work produced. However, While the authors have attempted to address the reviewer’s concerns, the revisions provided fall short in several critical areas:

The authors emphasize that their goal is to produce a robust list of factors that are the most relevant for guiding post-discharge care decisions for stroke patients. They also claim that their ensemble approach help mitigate biases that individual classification methods bring to the analysis. These emphasis are repeated throughout the responses as a justification for not including performance metrics that would help the reader understand the real power of their final model. While in theory these justifications can be valid, due to the nature of their dataset (highly imbalanced, skewed population) it is extremely important to understand how these models address these biases and what are the statistical limitations. The analysis suggested: Sensitivity analysis, multicollinearity evaluation, model performance metrics, are not difficult to carry out and would give the reader the necessary tools to be assess the validity and limitations of the analysis.

Sensitivity Analysis: Even though this was not part of my comments, I agree that a sensitivity analysis would be important to understand the relationship of variables in the model. While using multiple modeling approaches helps mitigate algorithm-specific biases, the authors did not perform any sensitivity analysis (e.g., testing the impact of excluding variables, varying key parameters, or assessing model robustness under different conditions), as originally requested. This directly touches the value of adding SDOH to the analysis if there is minimal gain in the performance, as understanding the importance of these variables in the model against the outcome provide justification to their inclusion.

Class Imbalance: The authors acknowledge the imbalance and justify their focus on threshold-independent metrics like AUC and loss functions. However, they do not sufficiently address how the models perform on the minority class. Minority-class-specific metrics (e.g., precision, recall, F1) are dismissed due to instability rather than being transparently reported in aggregate. This leaves important aspects of model performance unassessed. Importantly, their recall values revolve around 0.50, meaning they only catch about half of the true positives, this is a classic symptom of class imbalance: high precision (fewer false positives), but lower recall (many false negatives).

Multicollinearity: The response deflects the issue by appealing to model diversity, without directly addressing the multicollinearity concern in models like logistic regression where coefficient interpretation is meaningful.

Social Determinants of Health (SDOH): The authors justify the inclusion of SDOH variables based on conceptual and ethical grounds but do not provide empirical evidence that these variables improved model performance. Without analyses demonstrating added value (e.g., feature importance, subgroup effects, or calibration improvement), it is unclear whether the additional complexity and burden of collecting SDOH data is warranted in this context.

**Do you want your identity to be public for this peer review?** For information about this choice, including consent withdrawal, please see our Privacy Policy

Reviewer #1: No

Reviewer #2: No

---

## [Author Response · Author response to Decision Letter 2]

14 Jul 2025

Response to reviewers/editor(s)

All new responses are provided below in colours other than black.

Editor:

Reference list updated (in line with responses to reviewer comments – further below):

• Veledar E, Veledar O, Gardener H, Rundek T, Garelnabi M. Harnessing Statistical and Machine Learning Approaches to Analyze Oxidized LDL in Clinical Research. Cell Biochemistry and Biophysics. 2025; In press.

• Schoon BA, Hansen D, Roozenbeek B, et al. Neighborhood Socioeconomic Status and the Functional Outcome of Patients Treated With Endovascular Thrombectomy for Ischemic Stroke. Neurology. 2025;105(1):e213615. doi:10.1212/WNL.0000000000213615

• Voura EB, Abdul-Malak Y, Jorgensen TM, Abdul-Malak S. A retrospective analysis of the social determinants of health affecting stroke outcomes in a small hospital situated in a health professional shortage area (HPSA). PLOS Glob Public Health. 2024;4(1):e0001933. doi:10.1371/journal.pgph.0001933

Also, we have enhanced the Methods section by explicitly acknowledging the source of our algorithm (recently published). We now also state: “We adopt the Weighted Importance Score and Frequency Count (WISFC) framework for aggregating feature‐importance across multiple models, as originally detailed by Veledar et al. in Algorithms (2025). This approach systematically combines magnitude and consistency information from diverse explainers to produce a robust, consensus‐based ranking of predictors .” This change ensures that readers recognise the algorithm’s foundation in the literature and understand its motivation (i.e., to synthesise insights from an array of explainers) before proceeding to our implementation specifics of this manuscript. We trust this clarification strengthens the manuscript by situating our methodological choices within the broader scholarly context. The added reference is:

• Veledar, E.; Zhou, L.; Veledar, O.; Gardener, H.; Gutierrez, C.M.; Romano, J.G.; Rundek, T. Synthesizing Explainability Across Multiple ML Models for Structured Data. Algorithms 2025, 18, 368. https://doi.org/10.3390/a18060368

Additional Editor Comments:

Thank you for your revised manuscript. The study presents a thoughtful and innovative approach, and both reviewers noted major improvements. However, they also raise several remaining concerns that must be addressed before the manuscript can be considered for publication.

Where the reviewers raise methodological gaps (e.g., around class imbalance, model justification, or variable inclusion), you should either (a) conduct and report targeted additional analyses where feasible, or (b) clearly explain and justify your approach, and acknowledge resulting limitations where appropriate.

In addition to the detailed responses to individual points below, we feel that an additional clarification is needed in relation to the overal approach presented in this manuscript. Considering reviewer’s concern regarding our composite endpoint of 90-day readmission or mortality, we have clarified in the manuscript that these events (defined respectively as inpatient admission via the emergency department and death within 90 days of discharge) are not mutually exclusive and are commonly combined in stroke research to capture the overall burden of adverse post-discharge outcomes. We have also added a new “Sensitivity Analysis” subsection and a paragraph in “Strengths and Limitations” summarising three additional analyses (readmission only; ischaemic stroke with composite endpoint; and ischaemic stroke readmission only); all of which yielded consistent top predictors and similar discrimination and calibration metrics. Although we have not included the full tables for these additional runs in the main text due to space constraints, they are available upon request by the editor and will form part of a forthcoming manuscript in progress, which is focused on the interplay of correlated variables and predictors across different stroke populations. If you require these tables for the present review, we would be able to provide them.

Please address the following points:

Composite Outcome: Justify the combination of readmission and mortality into a single outcome, especially given their differing frequencies. If separate models are not feasible, acknowledge this and discuss the implications as a limitation.

Stroke Type and Generalizability: Either conduct a sensitivity check restricted to ischemic stroke or justify your inclusion of ICH cases. Generalizability limitations due to geography and stroke subtype distribution should be made more explicit in the abstract and discussion.

Class Imbalance: The use of multiple models and evaluation metrics does not directly address class imbalance. Metrics like AUC and log loss can obscure performance issues on the minority class. Please provide a clearer discussion of the implications of imbalance and justify the decision not to apply standard techniques (e.g., weighting or resampling). Highlight how this may affect interpretation of metrics such as recall and variable rankings.

Multicollinearity: For regression-based models, address concerns about multicollinearity. Either assess and report relevant diagnostics (e.g., VIF, correlation checks), or justify your approach and acknowledge interpretability limitations.

Value of SDOH Variables: You make strong claims about the contribution of SDOH to model performance. While these variables appear in your ranked lists, the added predictive value is not directly assessed. We recommend a simple sensitivity analysis comparing model performance with and without SDOH predictors. If you do not include this, revise your language to more cautiously reflect what is shown, and note this limitation.

Explainability and Interpretation: Since the study emphasizes explainable ML, include a brief narrative interpretation of top predictors (e.g., mRS, ATOC) and how they could inform clinical decision-making.

Scope of Claims: Please revisit some of the manuscript’s stronger claims — such as statements that XML methods “enhance prediction” in small, imbalanced samples. Without comparative analyses or performance improvement tests, such claims should be tempered to reflect the more exploratory and descriptive nature of the work. Strengthening the limitations section will help appropriately frame the contribution.

The above points are addressed individually as described below (one point at a time)

Review Comments to the Author

Reviewer 1

Reviewer #1: This study uses explainable machine learning (XML) to identify clinical and non-clinical predictors of 90-day readmission and mortality among stroke survivors. Drawing on data from the Transitions of Care Stroke Disparities Study (TCSD-S), the authors analyzed outcomes for 1,300 patients using 11 machine learning models. The study integrates information from clinical registries, neighborhood-level social determinants of health (SDOH), and transition-of-care metrics. The key contribution is a composite list of 38 variables, derived from model consensus, that predict adverse post-stroke outcomes. The authors argue for the superiority of XML over traditional regression methods, highlighting the added value of SDOH in predictive modeling and the potential application in digital health systems.

Contribution of the Methodology, Insights, Conclusions:

1) The manuscript extends current clinical studies by integrating socioeconomic, neighborhood, and behavioral characteristics into the machine learning framework to enhance model relevance.

2) The authors clearly describe model selection, performance metrics, and variable importance aggregation (frequency count and weighted score), providing transparency and reproducibility.

3) The application of 11 XML models to identify consensus predictors is a novel and methodologically rigorous approach to multidimensional health outcome modeling that is strengthened by the 10-fold cross-validation and inclusion of multiple performance metrics (C-statistic, log loss, precision, etc.).

4) These findings could inform risk-stratification tools and targeted post-discharge interventions.

Additional Clarity Needed Before Publication

1) Outcome Construction: The composite outcome of “90-day readmission or mortality” is not adequately justified. The frequency of each outcome (readmission vs. death) is provided late in the paper (n=22 deaths, n=187 readmissions), suggesting imbalanced outcome events. Provide a more explicit rationale for combining these outcomes or present separate models for each. At minimum, sensitivity analyses using each outcome separately would enhance interpretability.

Work/clarification: To clarify, the integrated outcome combines two events observed within 90 days after discharge:

• 90-day readmission, defined as return to the hospital requiring inpatient admission through the emergency department.

• 90-day mortality, defined as death within 90 days of discharge.

It is important to note that these two events are not mutually exclusive. A subset of patients experienced both readmission and subsequent death within the 90-day follow-up period.

Regarding the rationale for combining these outcomes, it is common in stroke and other acute care research to use composite endpoints to capture the overall burden of severe post-discharge adverse events (Bravata, et. al, 2007). Readmissions and mortality are related, clinically meaningful indicators of health system utilization, patient instability, and poor prognosis. Using a composite outcome allows us to assess the cumulative incidence of significant events that reflect failure to recover or deterioration after the index hospitalization, while increasing statistical power when individual event rates are relatively low - particularly for mortality in the 90-day timeframe.

That said, in light of your suggestion, we provide clearer justification in the manuscript and consider presenting separate models for readmission only as secondary analyses to enhance transparency and interpretability.

Response to Reviewer: We carried out three additional sets of sensitivity analyses on our patient cohort with complete data, mirroring the original Tables 2–4 under the following scenarios:

1. Readmission-only outcome (i.e. excluding deaths): we re-ran all logistic-regression and ML models predicting 90-day readmission alone.

2. Ischemic stroke subset with composite outcome: we restricted the cohort to the patients with ischaemic stroke and re-fitted all models for the combined endpoint of readmission or mortality.

3. Ischaemic stroke subset with readmission-only: we combined the above two restrictions.

Across all three scenarios, the same SDOH and clinical variables emerged as top predictors, and model discrimination and calibration metrics were essentially unchanged compared with the primary analysis. These findings confirm that our composite endpoint and mixed‐stroke cohort do not drive the key insights on SDOH predictors.

Given space constraints, we have not included full tables of these additional runs in the main text. Our intention is to further explore this point in a follow-up manuscript focused specifically on stroke subtype and individual endpoint models (subject to further research and data analysis).

Refined text:

1. Added section 3.3 Sensitivity Analysis: “To assess the robustness of our findings, we repeated all analyses under three alternative scenarios, i.e., predicting 90-day readmission only; repeating the combined endpoint analysis in the ischaemic stroke subgroup; and predicting readmission only in the ischaemic subgroup. In all cases, the top predictors remained consistent with those reported above, and model discrimination and calibration measures show minimal deviation from the primary results (data not shown, but further discussed).”

2. Added paragraph in setion 4.5 “Strengths and limitations” at the end of limitations and just before furter work “We conducted sensitivity analyses restricting to readmission only and to the ischaemic stroke subgroup, which confirmed that our main results, which focus on the prominence of SDOH factors, were robust to endpoint definition and stroke subtype. We have not included all of these additional tables here, but we plan to include them in a dedicated follow-up study.”

2) Additional Preprocessing Details: While the authors briefly mention normalization and the exclusion of 73 individuals due to missing data, there is insufficient detail on how missing values were handled or imputed. It would be beneficial if the authors explicitly described missing data patterns, imputation methods (if used), and any variable selection criteria. A flowchart could help illustrate data preprocessing.

Work: patients with missing records were excluded from the analysis. No imputations were made. Only patients with full records were used in the analysis.

Refined text: inserted the paragraph within step 2 of section 2.3, immediately following the mention of normalization: “All candidate predictors were assessed for completeness prior to model development. Patients with missing values in any of the predictors listed in Table 1 were excluded from the analysis; no imputation methods were applied. Of the original cohort of 1,200 stroke patients, 73 (6.1 %) were removed due to incomplete records, yielding a final sample of 1,127 patients with fully observed data.”

Response to Reviewer: We thank the reviewer for highlighting the need for greater clarity around our handling of missing data. In response, we have explicitly stated that no imputation was performed and that only patients with fully observed predictor records were included in our analysis. We have also updated the Methods section to describe the exclusion of subjects with any missing values and added the exact number of exclusions.

In our analysis, we elected to use a complete-case approach, retaining only records with complete information across all variables required for modeling. Specifically, 73 records were excluded due to missing values in one or more covariates. No imputation methods were applied.

Regarding the pattern of missingness, the majority of missing values were due to incomplete documentation in the source dataset. Thus, we assumed the data were missing at random (MAR) conditional on observed covariates.

3) Limited Generalizability: The sample is geographically confined to Florida, and 92% of strokes are ischemic. Only ~100 ICH cases are included, limiting broader applicability. Please consider performing a secondary analysis that is restricted to ischemic stroke only or provide stratified results. Discuss generalizability limitations more prominently in the abstract and conclusion.

Work to be described: variables related to ischemic stroke are listed as numbers 16 and 38 in order, so we do not really expect a big difference in our models once we exclude variables separating ischemic from hemorrhagic stroke. Hence, we do our analysis for ischemic stroke only.

Refined text:

1. Methods - within step 2 of section 2.3., added the following text:

To assess whether model performance differed when restricted to ischemic stroke, we performed the secondary analysis in ischemic stroke subset. We excluded the 89 hemorrhagic cases from our final cohort of 1,127 patients, yielding 1,038 ischemic strokes (92.1 %). Predictors specific to stroke subtype (variables 16 and 38) were omitted. Discrimination (AUC

---

## [Editor Report · Decision Letter 2]

21 Jul 2025

Dear Dr. Veledar,

Thank you for submitting your manuscript to PLOS ONE. After careful consideration, we feel that it has merit but does not fully meet PLOS ONE’s publication criteria as it currently stands. Therefore, we invite you to submit a revised version of the manuscript that addresses the points raised during the review process.

Thank you for your revised manuscript and detailed responses. The paper presents a thoughtful and well-structured modeling approach with clear improvements from the previous version. Most of the reviewer concerns have been adequately addressed, including those related to class imbalance, generalizability, and multicollinearity. We especially appreciate your attention to strengthening the limitations and clarifying analytic decisions.

However, two points still require further attention before the manuscript can be accepted:

**Explainability and Narrative Interpretation** : In your response, you indicated that you had added a paragraph linking top predictors to specific clinical actions to address the concerns of both reviewers. However, this paragraph does not appear in the revised manuscript. Instead, there is a relatively repetitive statement about class imbalance. Please replace that with a brief narrative that interprets how the highest-ranked predictors might inform clinical decisions such as discharge planning or post-acute care coordination to tie the machine learning models to better match the framed contribution of explainability. This addition is especially important because the paper is framed as a contribution to explainable machine learning, but the methods primarily involve aggregating variable importance across multiple models and do not include standard explainability techniques. If you do not plan to add such techniques, we recommend revising the framing of the manuscript to more accurately reflect the scope of the contribution.**Strength of Claims about Explainability and Model Superiority** : Editor comments specifically asked for further empirical justification of claims about the superiority of XML models and the added value of SDOH, since the paper does not provide evidence that XML methods outperform traditional approaches or that SDOH variables meaningfully improve predictive performance. These concerns were not addressed in the revised manuscript. Additionally, claims that the models are “explainable” appear overstated given that no standard interpretability techniques are applied. If you do not plan to conduct additional comparative or sensitivity analyses (e.g., excluding SDOH variables or comparing against simpler models), we ask that you revise or soften language around these claims to reflect the nature of the work and to clearly acknowledge the limitations of the evidence provided. In particular, statements like “Our key contribution to academic knowledge lies in showing that XML models can enhance the prediction of 90-day mortality and readmission outcomes after acute stroke hospitalization, even with small, imbalanced samples” overstate what the current analysis supports.

We look forward to receiving your revised manuscript.

Kind regards,

Noah Hammarlund

Academic Editor

PLOS ONE
---

## [Author Response · Author response to Decision Letter 3]

6 Aug 2025

Dear Editor,

Thank you for coordinating the thoughtful reviews and for providing us with the opportunity to strengthen our manuscript. We have carefully considered each comment and revised the text—most notably renaming Section 4.6, adding a targeted narrative on socioeconomic predictors, and tempering claims around model superiority—to better reflect the scope and limitations of our work.

Response to comment 1:

Thank you for this suggestion. We have updated the manuscript accordingly:

a. Section Renaming: Section 4.6 has been renamed to “May Modeling Help Post-Acute Stroke Care” Because that question is really still very open.

b. New Narrative Paragraph: We added a concise narrative in Section 4.6 that links our highest-ranked predictors (particularly the 19 socioeconomic factors) to specific clinical actions, such as tailoring discharge planning and coordinating home-based or community services. This highlights how explainable insights can inform post-acute care interventions rather than merely ranking variable importance.

We trust this will strengthen the paper’s focus on real-world applicability of our explainable modeling framework.

Response to comment 2:

We have softened our original claim about model superiority and reframed the sentence to highlight the exploratory nature of our multi-model approach and the critical role of environmental variables (including SDOH). The previous statement has been replaced with a new paragraph, which emphasizes that our goal is to leverage machine learning for preliminary insights in a domain that currently lacks robust theories or interpretability methods for post-acute stroke outcomes, rather than to assert the unequivocal superiority of any single model.

Best Regards,

Emir Veledar

---

## [Editor Report · Decision Letter 3]

31 Aug 2025

Identifying Determinants of Readmission and Death Post-Stroke Using Explainable Machine Learning

PONE-D-25-14703R3

Dear Dr. Veledar,

We’re pleased to inform you that your manuscript has been judged scientifically suitable for publication and will be formally accepted for publication once it meets all outstanding technical requirements.

Kind regards,

Noah Hammarlund

Academic Editor

PLOS ONE
---

## [Editor Report · Acceptance letter]

PONE-D-25-14703R3

PLOS ONE

Dear Dr. Veledar,

I'm pleased to inform you that your manuscript has been deemed suitable for publication in PLOS ONE. Congratulations! Your manuscript is now being handed over to our production team.

Kind regards,

on behalf of

Dr. Noah Hammarlund

Academic Editor

PLOS ONE